# Protoporphyrin IX (PpIX) Fluorescence during Meningioma Surgery: Correlations with Histological Findings and Expression of Heme Pathway Molecules

**DOI:** 10.3390/cancers15010304

**Published:** 2023-01-02

**Authors:** Dorothee C. Spille, Eva C. Bunk, Christian Thomas, Zeynep Özdemir, Andrea Wagner, Burak H. Akkurt, Manoj Mannil, Werner Paulus, Oliver M. Grauer, Walter Stummer, Volker Senner, Benjamin Brokinkel

**Affiliations:** 1Department of Neurosurgery, University Hospital Münster, Albert-Schweitzer-Campus 1, Building A1, 48149 Münster, Germany; 2Institute of Neuropathology, University Hospital Münster, 48149 Münster, Germany; 3Department of Radiology, University Hospital Münster, 48149 Münster, Germany; 4Department of Neurology with Institute of Translational Neurology, University Hospital Münster, 48149 Münster, Germany

**Keywords:** 5-aminolevulinic acid, ABCB6, ABCG2, CPOX, FECH, fluorescence, meningioma, protoporphyrin

## Abstract

**Simple Summary:**

In meningiomas, 5-aminolevulinc acid (5-ALA)-mediated fluorescence-guided resection (FGR) has been shown to improve intraoperative tumor bone and soft tissue invasion. However, several studies reported distinct limitations of FGR, e.g., for the visualization of the dura tail or CNS invasion. Notably, correlations between fluorescence and histological findings, as well as the expression of key heme synthesis pathway molecules, have been sparsely investigated. In this study, we examined 111 samples from 44 patients after an FGR of an intracranial meningioma for the presence of histopathological evidence of tumor tissue and intraoperative fluorescence, and analyzed the expression of key transporters/enzymes involved in PpIX metabolism using immunohistochemistry and qPCR. High sensitivity and specificity for the identification of tumor tissue and correlation of fluorescence and tumor tissue with expression of the enzymes/transporters were demonstrated. However, a deviating fluorescence and expression could be observed in non-neoplastic brain tissue, whereas this was lacking in the dura.

**Abstract:**

Background: The usefulness of 5-ALA-mediated fluorescence-guided resection (FGR) in meningiomas is controversial, and information on the molecular background of fluorescence is sparse. Methods: Specimens obtained during 44 FGRs of intracranial meningiomas were analyzed for the presence of tumor tissue and fluorescence. Protein/mRNA expression of key transmembrane transporters/enzymes involved in PpIX metabolism (ABCB6, ABCG2, FECH, CPOX) were investigated using immunohistochemistry/qPCR. Results: Intraoperative fluorescence was observed in 70 of 111 specimens (63%). No correlation was found between fluorescence and the WHO grade (*p =* 0.403). FGR enabled the identification of neoplastic tissue (sensitivity 84%, specificity 67%, positive and negative predictive value of 86% and 63%, respectively, AUC: 0.75, *p* < 0.001), and was improved in subgroup analyses excluding dura specimens (86%, 88%, 96%, 63% and 0.87, respectively; *p* < 0.001). No correlation was found between cortical fluorescence and tumor invasion (*p* = 0.351). Protein expression of ABCB6, ABCG2, FECH and CPOX was found in meningioma tissue and was correlated with fluorescence (*p* < 0.05, each), whereas this was not confirmed for mRNA expression. Aberrant expression was observed in the CNS. Conclusion: FGR enables the intraoperative identification of meningioma tissue with limitations concerning dura invasion and due to ectopic expression in the CNS. ABCB6, ABCG2, FECH and CPOX are expressed in meningioma tissue and are related to fluorescence.

## 1. Introduction

Meningiomas are the most common primary brain tumors and are usually treated through microsurgical resection in the case of space-occupying and/or symptomatic lesions. Numerous studies have reported the extent of resection, either in terms of the Simpson Classification system or the residual tumor volume on postoperative magnetic resonance imaging, as strong predictors for local tumor control [1,2]. Hence, the maximum safely achievable tumor resection is frequently strived for.

5-Aminolevulinic acid (5-ALA)-mediated fluorescence-guided resection (FGR) has been shown to improve the extent of resection and prognosis in high-grade gliomas [3] and, nowadays, is widely established in glioblastoma surgery [3,4]. In malignant cells, 5-ALA, a natural precursor of heme during hemoglobin biosynthesis, is intracellularly converted to protoporphyrin (Pp) IX through several enzymatic reactions. PpIX displays characteristic intraoperative fluorescence following excitation by blue–violet light (wavelength 375–440 nm), and, therefore, enables the identification of tumor issue with high specificity and sensitivity. Ultimately, PpIX is further converted to heme by the mitochondrial ferrochelatase (FECH) and underlies mitochondrial export [5]. Hence, the PpIX concentration depends on cellular and mitochondrial uptake, conversion to PpIX, as well as on further metabolism. Numerous studies have correspondingly reported correlations of tumor fluorescence with expression or activity of FECH or other key enzymes of the heme biosynthesis pathway (e.g., coproporphyrinogen-Oxidase, CPOX) or transmembrane transporters (e.g., ATP-binding cassette super-family B member 6, ABCB6 or ATP-binding cassette super-family G member 2, ABCG2) of PpIX in gliomas [6,7,8,9,10,11,12].

In meningiomas, the usefulness of FGR to improve tumor visualization and the extent of resection remains controversial [13,14,15]. In fact, 5-ALA-mediated tumor fluorescence is observed in the vast majority of meningiomas and has been reported to improve the visualization of tumor invasion into adjacent soft and bone tissue [13,14,15,16]. However, the utility of FGR to improve the visualization of dura invasion is unclear due to unsteady tumor fluorescence in the dura tail [16,17,18]. Fluorescence in the adjacent CNS tissue, as well as the utility of FGR during resection of recurrent meningiomas, e.g., to distinguish a tumor from scar tissue, remain to be further elucidated [15]. 5-ALA-mediated fluorescence in critical tissue invaded by or adjacent to meningiomas has, therefore, been analyzed in several studies using different devices and techniques, such as spectroscopy or spectrometry [15,17,18,19]. However, investigations about the expression of enzymes and transporters directly involved in PpIX metabolism in meningiomas and adjacent tissue are lacking, but are essential to further assess the usefulness of FGR in these lesions.

## 2. Materials and Methods

### 2.1. Patient Selection and Sample Collection

Forty-four patients who underwent fluorescence-guided surgery for primary diagnosed or recurrent intracranial meningioma in our department between October 2018 and February 2022 were included. FGR had been individually indicated according to the neurosurgeon’s preference, considering preoperative imaging and the medical history of the patient in each case. A weight-adjusted dose of 5-ALA (20 mg/kg body weight) was administered four hours prior to induction of anesthesia. Written informed consent was obtained from each patient. Surgery was then performed using a standard operating microscope (OPMI Pentero 900, Carl Zeiss AG, Oberkochen, Germany) equipped with a BLUE400 filter system. Postoperatively, the patient was protected from direct light exposition for 48 h. No 5-ALA-related side effects were registered. Intraoperatively, fluorescent and non-fluorescent specimens from the tumor and the surrounding tissue were taken by an experienced neurosurgeon. Intraoperatively, suspected diagnosis (e.g., tumor, dura, scar tissue) of the specimen was stated under white light as well as with the aid of neuronavigation. Subsequently, the surgeons switched to blue light and fluorescence was registered as present or absent. Samples of tumor tissue were collected from all patients. For ethical reasons, samples of non-tumoral tissue (scar, dura, brain, bone) were taken only from tissue in which the surgeon suspected tumor infiltration. The samples were, subsequently, subjected to standard neuropathological analyses using hematoxylin and eosin staining as well as elastic van Gieson staining. Subsequent diagnosis and grading were performed according to the WHO criteria for brain tumors of 2016 in all cases [20].

### 2.2. Immunohistochemistry

To analyze expression on a protein level, immunohistochemical staining for FECH, CPOX, ABCB6 and ABCG2 on formalin-fixed and paraffin-embedded meningioma tissue samples was performed using a standard protocol. In brief, 3–4 µm sections were deparaffinized and rehydrated through a graded alcohol series. After retrieval in sodium citrate buffer (pH = 6.0; Target Retrieval Solution S2369 DAKO; Agilent Technologies, Inc., Santa Clara, CA, USA; 1:10 diluted in distilled water) using the method of heat activation (40 min steam cooker, 20 min cooled at room temperature), staining was performed according to the manufacture’s protocol using an Agilent Autostainer Link 48 with a DCS DetectionLine (CEA1706) Kit. Staining the sections with DAB (3,3-diaminobenzidine)-Chromogene (DC135C006, Detection Kit and chromogen from DCS, Hamburg, Germany) was followed by a counterstaining process with hematoxylin. The sections were then dehydrated in an ascending series of alcohols (70%, 96%, 99%, Xylol) and finally sealed with Eukit and a cover slip for microscopic evaluation (Olympus BX-51). The applied antibodies, dilutions and controls are summarized in Table 1. To reduce interobserver variability, expression was examined by two independent observers who were blinded to the patient data with bright field microscopy for the presence and localization of FECH, CPOX, ABCB6 and ABCG2. For quantitative analyses, a semiquantitative score including the percentage of immunopositive tumor cells (0 ≤ 5%; 1 = 5–25%; 2 = 25–50%; 3 = 50–75%; 4 ≥ 75%) and the intensity of the staining (0 = no immunoreactivity; 1 = weak; 2 = moderate; 3 = strong staining) was established. The final score was then calculated by multiplying the staining intensity with density, and ranged from 0–12.

### 2.3. Quantitative Real-Time PCR (qPCR)

In 22 cases with available cryoasserved neoplastic and non-neoplastic tissue, relative expression of the genes of interest (FECH, CPOX, ABCB6 and ABCG2) on an mRNA level was analyzed with quantitative real-time PCR, using GAPDH as internal reference. RNA was extracted from the cells of the cryoasserved tissue using an RNA Isolation Kit (Maxwell 16 LEV simplyRNA Cell Kit AS1270, Promega, Mannheim, Germany) according to manufacturer’s instructions. Reverse transcription of RNA was performed using the High-capacity cDNA Reverse Transcription Kit (Applied Biosystems, Darmstadt, Germany), using 50 ng RNA as a template for the production of cDNA. Predesigned TaqMan Gene Expression Assays (Thermo Scientific, Dreieich, Germany, see Table 2) were used for qRT-PCR. Assays were used at least in triplicate according to the manufacturer’s instruction, using a StepOne real-time PCR system (Applied Biosystems). Relative expression of the genes of interest using GAPDH as internal reference were calculated with
ΔCt = (Ctgene of interest − CtGAPDH).

### 2.4. Statistical Analyses

For statistical analyses, standard commercial statistics software (IBM SPSS Statistics, Version 28, IBM, Ehningen, Germany) was used. Data are described by standard statistics with the median and range used for continuous variables and absolute and relative frequencies for categorical variables. Fisher’s exact and Mann–Whitney U tests were utilized for comparisons of categorical and continuous variables, respectively. The value of tissue fluorescence alone and of the surgeon’s impression—comprising information from white- and blue-light microscopy, neuronavigation and intraoperative in situ findings—for the prediction of meningioma tissue on microscopic slices was calculated using two-by-two contingency tables. Subsequently, the diagnostic value was quantified using the sensitivity and specificity, positive predictive value (PPV) and negative predictive value (NPV), and the area under the curve (AUC) was calculated using a receiver operating characteristic (ROC) analysis. A *p*-value of <0.05 was considered statistically significant throughout the analysis. All reported *p*-values are two-sided. Data collection and scientific use was approved by the local ethics committee in all cases (Ethik-Kommission der Ärztekammer Westfalen-Lippe und der Westfälischen Wilhelms-Universität, 2018-06-f-S).

## 3. Results

Table 3 summarizes the baseline clinical and histological data of the included 44 patients. Of those, 111 samples (mean: 2.5 samples per patient) were subjected to further analyses. Intraoperative fluorescence was observed in 70 cases (63%). In 76 samples containing meningioma tissue on microscopic analyses, fluorescence was absent in 21 cases (28%), and no correlation was found between fluorescence and the WHO grade of the tumor (60%, grade 1 vs. 69%, grade 2/3; *p* = 0.403). This also held true when exclusively analyzing 51 specimens from the bulk tumor (78% vs. 90%, *p* = 0.664).

### 3.1. Capability of FGR to Identify Neoplastic Tissue during Meningioma Surgery Varies

In the 70 specimens displaying intraoperative fluorescence, microscopic analyses displayed meningioma tissue in 56 cases, while no tumor was detected in 14 samples (80% vs. 20% *p* = 0.008). Thus, fluorescence alone predicted tumor tissue on microscopic slides with 73.1% sensitivity (95%CI 61.8–82.5%) and 59.4% specificity (95%CI 40.6–76.3%), and a PPV and NPV of 81.4% (95%CI 73.9–87.2%) and 47.5% (95%CI 26.3–59.0%), respectively. In subsequent ROC analyses, the prediction of tumor tissue with the use of intraoperative fluorescence alone showed an AUC value of 0.66 (95%CI 0.55–0.78, *p* = 0.006). On the other hand, in the 73 specimens suspected to contain a tumor by attending neurosurgeons using intraoperative white- and blue-light microscopy, neuropathological analyses revealed neoplastic meningioma tissue in 62 cases (85%). In contrast, meningioma tissue was found in 12 of 32 (36%) specimens intraoperatively suspected to be tumor-free. Thus, sensitivity and specificity of FGR for proper detection of meningioma tissue was 83.8% (95%CI 73.4–91.3%) and 66.7% (95%CI 47.2–82.7%), with a PPV of 86.1% (95%CI 78.7–91.2%) and an NPV of 62.5% (95%CI 48.4–74.8%, AUC: 0.75, 95%CI 0.64–0.87, *p* < 0.001).

Remarkably, in 17 specimens taken from the dura tail, tumor invasion was histopathologically confirmed in only 8 samples, and intraoperative fluorescence was only found in 3 of these cases (38%). Meningioma tissue was also found in two of seven dura specimens distant to the attachment of the main tumor (28%), and none of those samples displayed fluorescence. Thus, in the subgroup analyses excluding specimens taken from the dura, the sensitivity, specificity, PPV, NPV and AUC of FGR were 85.5% (95%CI 74.2–93.1%), 88.2% (95%CI 63.6–98.5%), 96.4% (95%CI 87.8–99.0%), 62.5% (95%CI 47.1–75.8%) and 0.87 (95%CI 0.77–0.97, *p* < 0.001), respectively.

In 22 samples from brain tissue adjacent to the bulk tumor, fluorescence was found in 16 (73%) but was absent in six cases (27%). Remarkably, among the first, neuropathological analyses revealed meningioma invasion in seven (44%) but non-neoplastic CNS tissue in nine cases (56%, *p* = 0.351). On the other hand, non-fluorescent samples displayed meningioma tissue in one case but non-neoplastic tissue in five cases. No correlation was found between the WHO grade of the tumor and cortical fluorescence (*p* = 1.00).

### 3.2. Protein Expression of ABCB6, ABCG2, FECH and CPOX

#### 3.2.1. Immunhistochemistry

At the protein level, observed using immunohistochemistry, the expression of the key PpIX transmembrane transporters as well as FECH and CPOX distinctly varied among the analyzed specimens and different tissues. In samples from the bulk tumor, the median expression scores for ABCB6, ABCG2, FECH and CPOX were 9 (range: 2–12), 8 (range: 3–12), 9 (range: 1–12) and 9 (range: 1–12), respectively, and expression was similar comparing grade 1 and 2/3 meningiomas (*p* > 0.05 for each). A strong relationship between intraoperative fluorescence and ABCB6 (*p* < 0.001), ABCG2 (*p* < 0.001), FECH (*p* < 0.001) and CPOX (*p* = 0.008) expression was also confirmed in cumulative analyzes of all samples (Figure 1).

Expression was lacking in tumor-free dura mater, but was present in the adjacent meningioma tissue (Figure 2). Notably, unspecific low-to-moderate expression of ABCB6 (median score 8, range: 8–9), ABCG2 (median score 3, range: 1–4), FECH (median score 4, range: 1–8) and CPOX (median score: 1, range: 0–1) was also found in all nine fluorescent cortex samples lacking tumor invasion (Figure 3). On the other hand, scar tissue was lacking ABCB6, FECH and CPOX expression (Figure 4). Occasionally, expression was also detected in non-fluorescent samples containing neoplastic tissue. Notably, the amount of meningioma tissue was considerably lower in these cases (Figure 5).

#### 3.2.2. Quantitative Real-Time PCR (qPCR)

At the mRNA level, qPCR revealed an increased expression of ABCB6 (3.95, 1.53–6.58), ABCG2 (4.54, 0.54–9.33), FECH (4.29, 1.65–7.15) and CPOX (5.57, 2.92–8.44) (ΔCt median and range, each) in the 22 available samples. However, statistical analyses showed a similar median ΔCt of all analyzed enzymes and transporters in fluorescent and non-fluorescent samples (Figure 6).

## 4. Discussion

Similar to previous studies, tumor fluorescence in our study was not correlated with the WHO grade of the lesion [15,19]. We further found that the capability of tissue fluorescence alone to predict meningioma tissue in microscopic analyses was only moderate, with distinct limitations concerning specificity. However, the capability was considerably increased for FGR, which integrates the surgeon’s impressions from both blue- and white-light microscopy with intraoperative findings, such as anatomical information and neuro-navigation data. Similar results were recently reported by Wadiura et al., who found a distinctly higher PPV of fluorescence in specimens taken from the bulk tumor than from the peritumoral tissue [21]. Therefore, while reflecting the capability of fluorescence alone and FGR to improve the intraoperative visualization of tumor tissue, this finding also underlines that a critical evaluation of fluorescent but doubtful neoplastic tissue prior to resection is mandatory.

It was of note that the fluorescence at the dura tail was visible in <50% of specimens, despite microscopic evidence of meningioma tissue, and that tumor tissue was present in two of seven non-fluorescent dura specimens distant to the bulk tumor. A lack of fluorescence in the dura tail despite microscopic evidence of meningioma tissue has been shown in several studies [16,22,23]. Spectrometric analyses of our group and other groups suggest that the amount of tumor tissue, but not the lack of PpIX accumulation, contributes to the missing intraoperatively visible fluorescence of the dura tail [17,18]. This finding was also supported by our current results, displaying a brisk expression of PpIX transporters and enzymes of the heme pathway in fluorescent and non-fluorescent specimens from the dura tail. Correspondingly, the capability of FGR to predict microscopic evidence of meningioma tissue was distinctly higher when excluding dura samples. Notably, the portion of immunopositive neoplastic tissue in non-fluorescent samples was considerably lower, further indicating that the amount of meningioma tissue rather than the expression of PpIX transporters and metabolism enzymes contributes to intraoperative visible fluorescence. To overcome the surgeon’s subjectivity and intraoperative confounding factors (light sources, photo bleaching, microscope performance), objective quantification should be conducted. There have been studies on image quality and diagnostic accuracy using a confocal laser microscope ex vivo and in vivo [24], and using a multimodal two-photon fluorescence endomicroscope [25], in order to differentiate tumoral from non-tumoral tissue in meningioma patients.

Cortical/arachnoidal fluorescence in our study was unspecific and not related to tumor invasion. Similar data were provided by Wadiura et al., who reported an absence of tumor tissue in six of seven fluorescent cortical specimens obtained during meningioma surgery [22]. However, as the first study to do so thus far, our results suggested an underlying aberrant expression of key molecules of the heme pathway in these cases. In contrast, Cornelius et al. detected meningioma invasion in all fluorescent specimens taken from the adjacent cortex [26]. In gliomas, Lau et al. observed fluorescence in 35% of their intraoperative biopsies from tumor-free brain tissue [27]. Thus, the utility of FGR to visualize brain invasion during meningioma surgery remains to be further investigated.

Despite numerous studies on gliomas, the expression of transmembrane transporters and enzymes involved in PpIX metabolism in meningiomas remains largely unexplored. As expected, most tumor samples displayed expression of ABCB6, ABCG2, FECH and CPOX in our study, and expression was strongly correlated with fluorescence.

ABCB6 is a transmembrane transporter that enables mitochondrial coproporphyrinogen III influx during PpIX and heme biosynthesis. Correspondingly, increased expression was shown to induce both PpIX accumulation and susceptibility to 5-ALA-based photodynamic therapy (PDT) in gliomas [12].

In contrast, the transmembrane transporter ABCG2 induces mitochondrial efflux of PpIX and is expressed, e.g., in malignant gliomas, breast and colorectal cancer [7,28,29]. An upregulation of ABCG2 was further shown to reduce the efficacy of PDT in brain tumors [6], while its inhibition increased both intracellular PpIX levels and PDT efficacy [9]. As the only study on meningiomas, Freitag et al. reported similar ABCG2 expression in dura and neoplastic tissue, hence contradicting our findings from immunohistochemistry and qPCR [30].

FECH catalyzes the insertion of iron into PpIX, leading to a reduction in intracellular PpIX. Correspondingly, a downregulation of FECH is considered to induce PpIX accumulation and tumor fluorescence in malignant gliomas [11]. In 2010, Hefti et al. showed FECH activity in grade 1 meningioma cell lines and further revealed a correlation with susceptibility to 5-ALA-based photodynamic therapy [31]. Similar to ABCG2, increased expression of FECH in fluorescent tissue in our study, therefore, appears to be contradictory, but presumably rather reflected PpIX metabolism in the analyzed tissue in general.

CPOX catalyzes oxidation of coproporphyrinogen III to PpIX, and upregulation was shown in malignant gliomas and correlated with intraoperative tumor fluorescence [8,10], which matched our observations in meningiomas.

Notably, qPCR revealed an increased mRNA expression of FECH, CPOX, ABCB6 and ABCG2 in the analyzed samples. However, in contrast to protein expression, mRNA expression was not related to fluorescence. Interestingly, discrepancies between the expression of key heme pathway molecules at the mRNA and protein levels, as well as varying correlations with fluorescence, have recently been reported in gliomas by Mischkulnig et al. [32]. These findings might be explained by an intermediary mechanism between transcription and translation involved in PpIX fluorescence in both gliomas and meningiomas. Furthermore, mRNA levels are a reflection of the average gene expression in the entire FFPE slice. Sparsely sampled tissue material and suboptimal RNA quality can be a cause of a decrease in genes detected. This may impede the quantification and offers advantages for an image analysis-assisted scoring method, such as immunohistochemistry, to provide improved results.

The authors are aware of some limitations of the study. In fact, intraoperative assessment of fluorescence as well as sampling by the surgeon were not standardized and prone to subjectivity and sampling errors. For ethical reasons, only suspected tumor tissue was removed, which in turn may have resulted in a selection bias. Although ALA/PpIX uptake and metabolism impacts the visibility of intraoperative fluorescence [33,34], the exact duration between ALA administration and sampling was not recorded.

## 5. Conclusions

In conclusion, the diagnostic capacity of FGR to identify tumor tissue intraoperatively is excellent but suffers limitations when detecting tumor invasion in dura and CNS tissue. Expression of the analyzed key PpIX/heme pathway molecules was mostly limited to neoplastic tissue and strongly correlated with fluorescence. However, aberrant expression in CNS tissue was occasionally found and might substantially contribute to deviant fluorescence and should be considered during intraoperative decision-making and when interpreting results of previous and future studies about FGR in meningiomas.

## Figures and Tables

**Figure 1 cancers-15-00304-f001:**
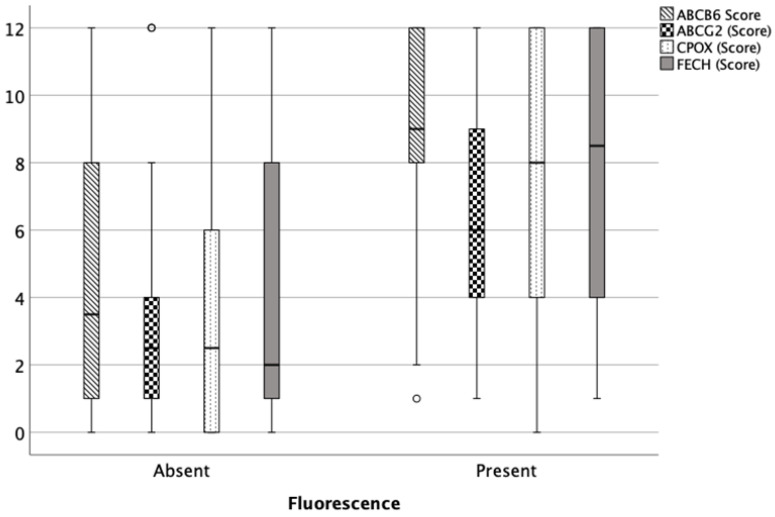
Box-and-whisker plots illustrating ABCB6, ABCG2, FECH and CPOX protein expression correlated with fluorescence, determined with immunohistochemical staining. Median expression scores from immunohistochemistry of all four proteins were significantly higher in fluorescent than in non-fluorescent specimens (*p* < 0.001, each). The boxes indicate upper and lower 25% quartiles, the whiskers indicate the minimum and maximum value, the dots indicate the outliers, and the heavy horizontal line indicates the median.

**Figure 2 cancers-15-00304-f002:**
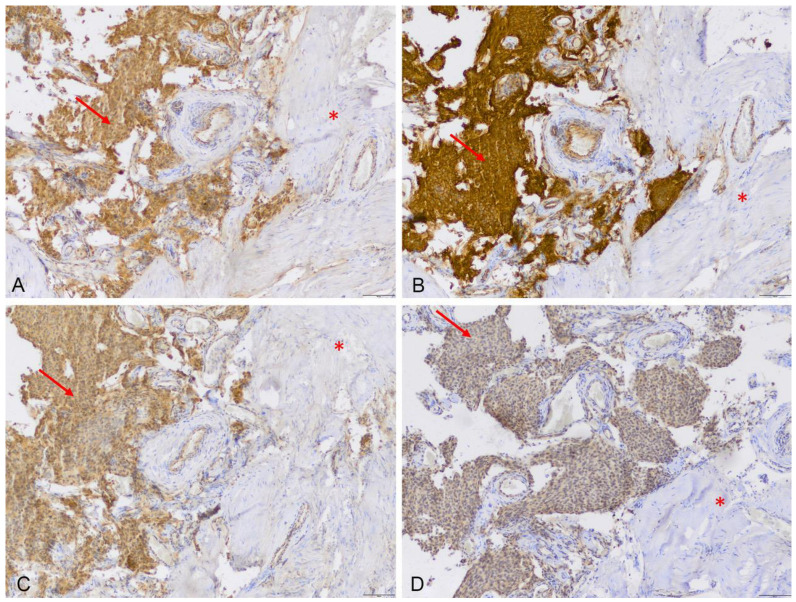
Illustrative samples of expression of the analyzed transporters and metabolism enzymes in the dura attachment of a meningioma, taken using immunohistochemistry. Expression of ABCB6 (**A**), ABCG2 (**B**), FECH (**C**) and CPOX (**D**) was found in tumor tissue (arrows) but not in the adjacent dura mater (*, scale bar 100 µm, magnification 10-fold).

**Figure 3 cancers-15-00304-f003:**
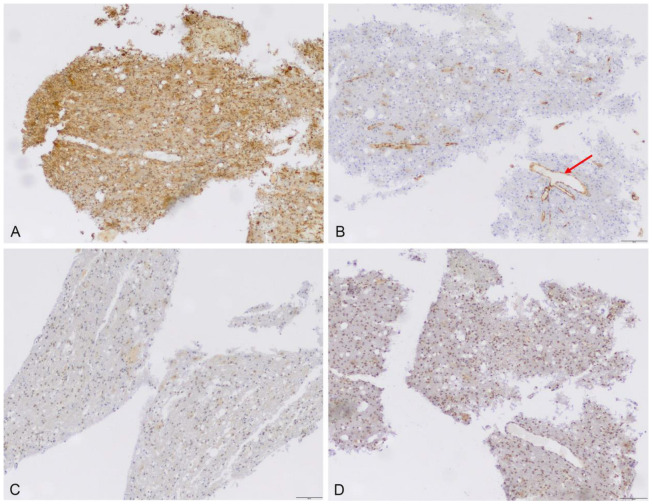
Protein expression of ABCB6, ABCG2, FECH and CPOX in fluorescent non-neoplastic CNS tissue. Both diffuse ABCB6 (**A**) and nuclear CPOX (**D**) as well as weak nuclear FECH (**C**) and patchy ABCG2 (**B**) protein staining could be detected. The arrow in (**B**) indicates endothelial ABCG2 staining (scale bar 100 µm, magnification 10-fold).

**Figure 4 cancers-15-00304-f004:**
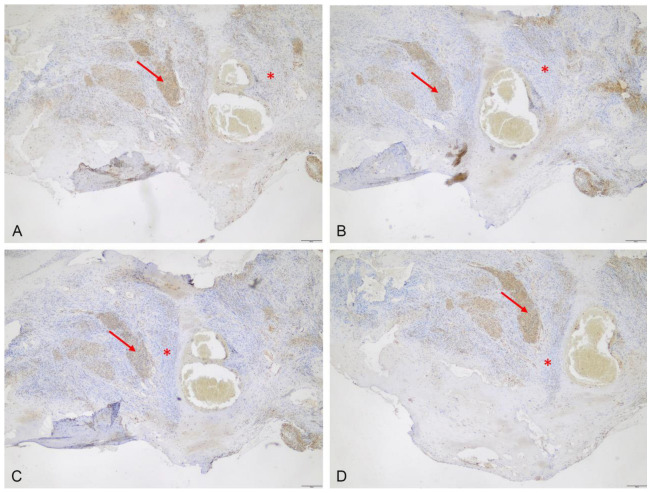
Expression of the analyzed transporters and metabolism enzymes in scar tissue. Protein expression of ABCB6 (**A**), ABCG2 (**B**), FECH (**C**) and CPOX (**D**) is absent in scar tissue (*) but present in the invading meningioma tissue (arrows, scale bar 200 µm, magnification 5-fold).

**Figure 5 cancers-15-00304-f005:**
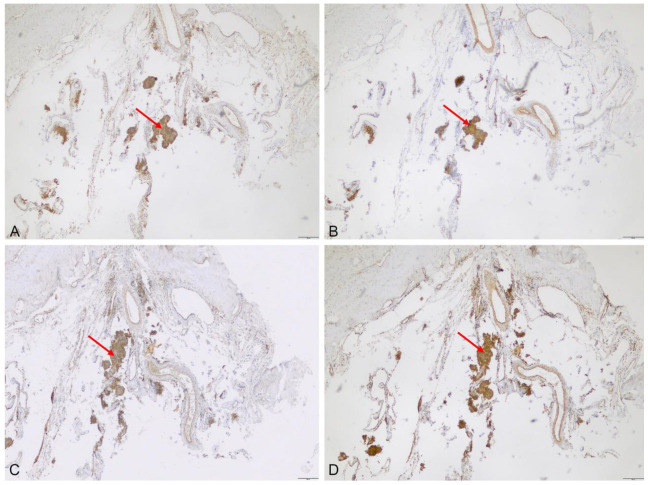
Expression of the analyzed transporters and metabolism enzymes in a non-fluorescent sample. Although strong expression of ABCB6 (**A**), ABCG (**B**), FECH (**C**) and CPOX (**D**) was found in nests of meningioma tissue (arrows), the amount of tumor in the analyzed sample was considerably lower (scale bar 200 µm, magnification 5-fold).

**Figure 6 cancers-15-00304-f006:**
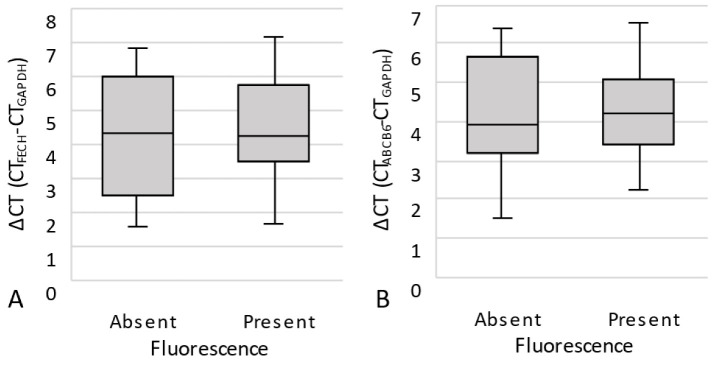
Box-and-whisker plots illustrating ABCB6, ABCG2, FECH and CPOX mRNA expression correlated with fluorescence, determined using qPCR. No correlation of the relative expression of ABCB6 ((**A**), *p* = 0.972), ABCG2 ((**B**), *p* = 0.896), FECH ((**C**), *p* = 0.744) and CPOX ((**D**), *p* = 0.393) with tumor fluorescence at the mRNA level was found. The boxes indicate upper and lower 25% quartiles, the whiskers indicate the minimum and maximum value, the dots indicate the outliers, and the heavy horizontal line indicates the median.

**Table 1 cancers-15-00304-t001:** Summary of the antibodies and controls used for immunohistochemical staining.

Antigene	Manufacturer	Order#	Dilution	Positive Control	Host Species
ABCB6	Invitrogen	PA5-78693	1:100	Mamma carcinoma	Rabbit
CPOX	Invitrogen	PA5-97613	1:200	Liver	Rabbit
FECH	Abcam	ab219349	1:288	Kidney	Rabbit
ABCG2	Abcam	ab229193	1:2000	Placenta	Rabbit

**Table 2 cancers-15-00304-t002:** Summary of the gene expression assays used for qPCR. Predesigned TaqMan Gene Expression Assays (Thermo Scientific, Dreieich, Germany) were used to quantify ABCB6, ABCG2, CPOX and FECH mRNA expression in relation to the internal control (GAPDH).

Gene of Interest	TaqMan Assay (Thermo Fisher)
ABCB6	Hs00180568-m1
CPOX	Hs01071019-m1
FECH	Hs00164616-m1
ABCG2	Hs01053790-m1
GAPDH (Reference)	Hs02786624-g1

**Table 3 cancers-15-00304-t003:** Baseline clinical and histological characteristics. Samples included both tumoral and peritumoral tissue. PRRT = Peptide receptor radionuclide therapy.

Variable	N	(*n*%)
Age (median)	61 years (19–85)	
Females	27	61
Males	17	39
Tumor location		
Convexity/parasagittal	34	77
Skull base	10	23
Indication		
Primary diagnosis	36	82
Recurrence	8	18
Previous therapy		
Previous microsurgery	1	2
Previous microsurgery and irradiation	4	9
Previous irradiation	1	2
Previous microsurgery, irradiation and PRRT	1	2
Extent of resection		
Simpson I–III	37	84
Simpson IV–V	7	16
WHO Grade		
1	37	84
2/3	7	16

## Data Availability

Not applicable.

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
