# Peer review of "Protoporphyrin IX (PpIX) Fluorescence during Meningioma Surgery: Correlations with Histological Findings and Expression of Heme Pathway Molecules"

_cancers, 2023, doi:10.3390/cancers15010304_

Round 1
Reviewer 1 Report
The authors present a retrospective analysis of 44 patients that were operated using fluorescence-guided resection. The ability of fluorescence to predict WHO grade and invasive growth was assessed. Tumor tissue was analyzed for key transporters and enzymes involved in PpIX metabolism in the resected tissue. The introduction gives a nice overview and prepares the reader well for the subject. Materials and methods seem complete. The results are well presented and the discussion is critical and focused. A few questions arose that I would like the authors to address:
1. Positive fluorescence intraoperatively was seen in 70 of 111 samples from 44 intracranial meningiomas. It would be of interest to know how the rate of positive fluorescence among the 44 meningiomas was and how the number of samples were chosen for each tumor. The authors state in materials and methods, that the tissue sampling was done according to the neurosurgeon’s preferences. Maybe this could be described in a bit more detail. Was there a certain agenda for each surgeon to provide tissue of definite tumor and tissue questionable whether it was tumor or not? This would be important to know since the number of samples and the way tissue was chosen clearly influence the resulting specificity and sensitivity.
2. Figures 2 – 5: The scale bars are very small and hard to read.
3. The discrepancy between the expression results of immunohistochemistry and qPCR could be further discussed. Immunohistochemical scoring can always be criticized, but the authors have applied a quite detailed semiquantitative scoring method, which I believe is sufficient but should be discussed.
4. It is also quite astonishing, that merely 85% of samples chosen by the surgeon as tumor, contained meningioma tissue. Furthermore, it would be interesting what these 15% of samples contained other than tumor (Scar tissue? Cortex?). This underlines how difficult it can be to differentiate tissue intraoperatively and the need to further develop the surgeon’s armamentarium of intraoperative tools to do so. Regarding the current manuscript, the authors have provided proof that FGR may not be of big help. But what do the authors think about other future directions?
Author Response
Title of the Manuscript: Protoporphyrin IX (PpIX) fluorescence during meningioma surgery: Correlations with histological findings and expression of heme pathway molecules
Manuscript Number: cancers-2108412
We would like to take this opportunity to thank the reviewers and the editor for the constructive comments made about our work and also thank the editorial board for giving us an opportunity to resubmit our work to “Cancers”.
We have made additional changes in the manuscript to address the reviewers’ concerns and hope that our responses to questions and the modifications to our manuscript are sufficient for publication in “Cancers“.
Reviewer 1:
The authors present a retrospective analysis of 44 patients that were operated using fluorescence-guided resection. The ability of fluorescence to predict WHO grade and invasive growth was assessed. Tumor tissue was analyzed for key transporters and enzymes involved in PpIX metabolism in the resected tissue. The introduction gives a nice overview and prepares the reader well for the subject. Materials and methods seem complete. The results are well presented and the discussion is critical and focused. A few questions arose that I would like the authors to address:
- Positive fluorescence intraoperatively was seen in 70 of 111 samples from 44 intracranial meningiomas. It would be of interest to know how the rate of positive fluorescence among the 44 meningiomas was and how the number of samples were chosen for each tumor. The authors state in materials and methods, that the tissue sampling was done according to the neurosurgeon’s preferences. Maybe this could be described in a bit more detail. Was there a certain agenda for each surgeon to provide tissue of definite tumor and tissue questionable whether it was tumor or not? This would be important to know since the number of samples and the way tissue was chosen clearly influence the resulting specificity and sensitivity.
Authors’ response: Thank you very much for this useful advice. Of the 44 patients, 6 showed no fluorescence intraoperatively after ALA administration, despite sampling from tumor tissue.
It is correctly pointed out by the reviewer that to date no consented criteria for testing the diagnostic accuracy of fluorescence guided resection exist resulting in lack of comparability and reproducibility of methods.
The samples were taken by surgeons who are very experienced in performing fluorescence-assisted resections. For this purpose, a specific tissue (e.g. tumor, dura, scar) was first identified under white light as well as with the aid of neuronavigation. Subsequently, the surgeons switched to blue light to detect the presence or absence of fluorescence and obtain a biopsy. Samples of tumor tissue were collected from all patients. For ethical reasons, samples of non-tumoral tissue were taken only from tissue in which the surgeon was unsure of tumor infiltration. This concerns dura, brain tissue, bone and scar tissue. Quite correctly, this is accompanied by a certain selection bias, but this can hardly be avoided.
Change to text: Intraoperatively, fluorescent and non-fluorescent specimen from the tumor and the surrounding tissue were taken by an experienced neurosurgeon. Intraoperatively suspected diagnosis (e.g. tumor, dura, scar tissue, etc.) of the specimen was stated under white light as well as with the aid of neuronavigation. Subsequently, the surgeons switched to blue light and fluorescence was registered dichotomized as present or absent. Samples of tumor tissue were collected from all patients. For ethical reasons, samples of non-tumoral tissue were taken only from tissue in which the surgeon suspected tumor infiltration. (page 3, paragraph 2, lines 49f. – page 4, paragraph 1, lines 1-5)
- Figures 2 – 5: The scale bars are very small and hard to read.
Authors’ response: We completely agree with the reviewer. However, the program used for microscopic photo acquisition inserts the scales automatically. Thus, it is technically difficult to magnify without distorting the actual range of values. Therefore, we have additionally indicated the magnification under the figures.
- The discrepancy between the expression results of immunohistochemistry and qPCR could be further discussed. Immunohistochemical scoring can always be criticized, but the authors have applied a quite detailed semiquantitative scoring method, which I believe is sufficient but should be discussed.
Authors’ response: Thank you for the perceptive comment. An explanation for the discrepancy between the expression results of immunohistochemistry and qPCR may be that mRNA levels are a reflection of the average gene expression in the entire FFPE slice, whereas immunohistochemistry is more distinguish in favor of selected representative tumor areas. In our study, samples were collected with little tissue material in some cases. This may impede quantification and offers advantages for an image analysis-assisted scoring method such as immunohistochemistry to provide distinguished results. Furthermore, RNA integrity is critical for determining gene expression, and suboptimal RNA quality can be a cause of this decrease in genes detected. Since the tissue was cryoasserved and subsequently thawed to perform qPCR, this may be a methodological error. We have now addressed this issue in the manuscript.
Change to text: Furthermore, mRNA levels are a reflection of the average gene expression in the entire FFPE slice. Sparely sampled tissue material and suboptimal RNA quality can be a cause of a decrease in genes detected. This may impede quantification and offers advantages for an image analysis-assisted scoring method such as immunohistochemistry to provide distinguished results. (page 12, paragraph 1, lines 31-36)
- It is also quite astonishing, that merely 85% of samples chosen by the surgeon as tumor, contained meningioma tissue. Furthermore, it would be interesting what these 15% of samples contained other than tumor (Scar tissue? Cortex?). This underlines how difficult it can be to differentiate tissue intraoperatively and the need to further develop the surgeon’s armamentarium of intraoperative tools to do so. Regarding the current manuscript, the authors have provided proof that FGR may not be of big help. But what do the authors think about other future directions
Authors’ response: Nine biopsies compromised dura tissue without tumor infiltration and one sample contained scar tissue. The expression of the transporters/ enzymes in tumor tissue, whereas expression is absent in Dura, supports the benefit of using ALA-guided resection in meningioma patients. One approach would be to use objective quantification instead of the surgeon's subjective assessment of whether fluorescence is visible. The use of mass spectrometry or a confocal endomicroscope could be further investigated. In this regard, our research group and other colleagues have published promising results, which we have added to our manuscript.
Change to text: To overcome the surgeon's subjectivity and intraoperative confounding factors (light sources, photo bleaching, microscope performance), objective quantification should be conducted. There are studies on image quality and diagnostic accuracy using a confocal laser microscope ex vivo and in vivo (Xu Y, Abramov I, Belykh E, Mignucci-Jiménez G, Park MT, Eschbacher JM, Preul MC. Characterization of ex vivo and in vivo intraoperative neurosurgical confocal laser endomicroscopy imaging. Front Oncol. 2022 Aug 24;12:979748. doi: 10.3389/fonc.2022.979748) or a multimodal two-photon fluorescence endomicroscope (Mehidine H, Refregiers M, Jamme F, Varlet P, Juchaux M, Devaux B, Abi Haidar D. Molecular changes tracking through multiscale fluorescence microscopy differentiate Meningioma grades and non-tumoral brain tissues. Sci Rep. 2021 Feb 15;11(1):3816. doi: 10.1038/s41598-020-78678-4.)in order to differentiate tumoral from non-tumoral tissue in meningioma patients. (page 11, paragraph 1, lines 27-32)
- Reviewer:
- The authors investigate the expression of enzymes and transporters involved into PpIX metabolism in meningiomas and adjacent tissue (bone and dural tail), in order to assess the usefulness of fluorescence-guided surgery also during meningioma’s removal. In the preface of this article, the cited references about the relationship between the PpIX metabolism and meningiomas are relevant, but maybe not updated with the last analyses, both in vivo and in vitro, available in literature.
Authors’ response: Thank you for pointing this out. We have added studies on future solution approaches.
Change to text: There are studies on image quality and diagnostic accuracy using a confocal laser microscope ex vivo and in vivo (Xu Y, Abramov I, Belykh E, Mignucci-Jiménez G, Park MT, Eschbacher JM, Preul MC. Characterization of ex vivo and in vivo intraoperative neurosurgical confocal laser endomicroscopy imaging. Front Oncol. 2022 Aug 24;12:979748. doi: 10.3389/fonc.2022.979748) or a multimodal two-photon fluorescence endomicroscope (Mehidine H, Refregiers M, Jamme F, Varlet P, Juchaux M, Devaux B, Abi Haidar D. Molecular changes tracking through multiscale fluorescence microscopy differentiate Meningioma grades and non-tumoral brain tissues. Sci Rep. 2021 Feb 15;11(1):3816. doi: 10.1038/s41598-020-78678-4.)in order to differentiate tumoral from non-tumoral tissue in meningioma patients. (page 11, paragraph 1, lines 27-32)
- The study design and methods are described in an appropriate way. In Materials and Methods, from lines 94 to 97, the authors declare: “Intraoperatively, fluorescent and non-fluorescent specimen from the tumor and the surrounding tissue were taken according to the neurosurgeon’s preference, and fluorescence was registered dichotomized as present or absent.” I suggest to better specify the surgical strategy used for the specimen’s collection, in order to methodologically improve the study description. Tumor and the surrounding tissue, and then the collected specimens into and outside the tumor, were evaluated based only on the surgeon’s impression and expertise, or the boundary between them was also evaluated with the aid of neuronavigation system?
Authors’ response: Yes, indeed, this is an important point, we used a neuronavigation system (Brain Lab software) to select the tissue samples. We have included this in the methods section.
Change to text: Intraoperatively suspected diagnosis (e.g. tumor, dura, scar tissue, etc.) of the specimen was stated under white light as well as with the aid of neuronavigation. (page 3, paragraph 2, line 50 – page 4, paragraph 1 line 1)
- What exactly were the areas sampled in each patient (e.g. tumor core; tumor periphery; outside the tumor)?
Authors’ response: One sample was taken from the main tumor of each patient and additional samples, if tumor infiltration was suspected, were taken from the surrounding tissue (scar, dura, dura tail, brain tissue). The method section was adapted accordingly.
Change to text: Samples of tumor tissue were collected from all patients. For ethical reasons, samples of non-tumoral tissue (scar, dura, brain, bone) were taken only from tissue in which the surgeon suspected tumor infiltration. (page 4, paragraph 1, lines 2-5)
- For each patient, how many samples were obtained? Has the number of lesion samples been standardized?
Authors’ response: Thank you for the constructive feedback. At least one sample was taken from each patient (mean: 2.5 samples per patient). Depending on the intraoperative conditions, the surgeons weighed how many additional samples should be taken. We are aware of the lack of standardization in sample collection, but it was done this way for ethical reasons. We have addressed the resulting selection bias in the limitations section.
Change to text: For ethical reasons, only tumor-suspect tissue was removed, which in turn may result in selection bias. (page 12, paragraph 1, lines 39f.)
- In the evaluation of the fluorescence, not always is possible dichotomizing in present or absent. How were the intermediate (weak) fluorescence conditions evaluated, in order to make the fluorescent definition as objective as possible?
Authors’ response: This is a very observant note. Indeed, there was weak/ intermediate fluorescence. This was finally considered by us to be positive. We did this consciously against the background that the surgeon must also decide for or against resection in the intraoperative setting and has no "interim solution".
However, the results of the study, in particular the statistical analysis and the immunohistochemical results have been clearly presented.

Reviewer 2 Report
The authors investigate the expression of enzymes and transporters involved into PpIX metabolism in meningiomas and adjacent tissue (bone and dural tail), in order to assess the usefulness of fluorescence-guided surgery also during meningioma’s removal. In the preface of this article, the cited references about the relationship between the PpIX metabolism and meningiomas are relevant, but maybe not updated with the last analyses, both in vivo and in vitro, available in literature.
The study design and methods are described in an appropriate way. In Materials and Methods, from lines 94 to 97, the authors declare: “Intraoperatively, fluorescent and non-fluorescent specimen from the tumor and the surrounding tissue were taken according to the neurosurgeon’s preference, and fluorescence was registered dichotomized as present or absent.” I suggest to better specify the surgical strategy used for the specimen’s collection, in order to methodologically improve the study description.
1. Tumor and the surrounding tissue, and then the collected specimens into and outside the tumor, were evaluated based only on the surgeon’s impression and expertise, or the boundary between them was also evaluated with the aid of neuronavigation system?
2. What exactly were the areas sampled in each patient (e.g. tumor core; tumor periphery; outside the tumor)?
3. For each patient, how many samples were obtained? Has the number of lesion samples been standardized?
4. In the evaluation of the fluorescence, not always is possible dichotomizing in present or absent. How were the intermediate (weak) fluorescence conditions evaluated, in order to make the fluorescent definition as objective as possible?
However, the results of the study, in particular the statistical analysis and the immunohistochemical results have been clearly presented.
Author Response

(The authors gave the same response as above.)

Round 2
Reviewer 2 Report
I appreciate the effort of the authors to answer the reviewers' questions and implement the text with the required details, particularly in the materials and methods section.
I confirm the opinion that unfortunately the lack of standardization in the samples collection and the dichotomization of fluorescence in "present/absent", continue to represent a limitation in this study.
However, the developed idea and the research line are interesting and I believe that the article deserves to be published.